# The Effects of Tinnitus on Significant Others

**DOI:** 10.3390/jcm11051393

**Published:** 2022-03-03

**Authors:** Eldre Wiida Beukes, Alyssa Jade Ulep, Gerhard Andersson, Vinaya Manchaiah

**Affiliations:** 1Vision and Hearing Sciences Research Group, Anglia Ruskin University, Cambridge CB1 1TP, UK; 2Virtual Hearing Lab, Collaborative Initiative between University of Colorado School of Medicine and University of Pretoria, Aurora, CO 80045, USA; vinaya.manchaiah@cuanschutz.edu; 3Department of Speech and Hearing Sciences, Lamar University, Beaumont, TX 77705, USA; aulep@lamar.edu; 4Department of Behavioral Sciences and Learning, Linköping University, 58183 Linköping, Sweden; gerhard.andersson@liu.se; 5Department of Clinical Neuroscience, Division of Psychiatry, Karolinska Institute, 17177 Stockholm, Sweden; 6Department of Otolaryngology—Head and Neck Surgery, University of Colorado School of Medicine, Aurora, CO 80045, USA; 7Department of Speech—Language Pathology and Audiology, University of Pretoria, Gauteng 0001, South Africa; 8Department of Speech and Hearing, Manipal College of Health Professions, Academy of Higher Education, Manipal 576104, India

**Keywords:** significant others, third-party disability, tinnitus, positive experiences, intervention

## Abstract

Although chronic conditions could cause third-party disability for significant others (SOs), little is known regarding the impact of tinnitus on SO. This study aimed to identify the effects of tinnitus on SOs. SOs of individuals with tinnitus were invited to participate in this study. SOs completed three open-ended questions focusing on the effects of tinnitus. Individuals with tinnitus completed the Tinnitus Functional Index as a self-reported measure of tinnitus severity. A mixed-methods analysis approach was undertaken. Of the 156 SOs responding, 127 (85%) reported that tinnitus impacted them. The impact surrounded sound adjustments, activity limitations, additional demands, emotional toll, and helplessness. Tinnitus negatively affected the relationship for 92 (58%) due to communication frustrations and growing apart. When asked if tinnitus had any positive effects, 64 (47%) SOs reported positive lifestyle adaptions, personal development, health awareness, and a changed outlook. There was no association between the level of tinnitus severity and SOs reporting that tinnitus had an impact on them individually, their relationships, or those reporting positive experiences. The study highlighted the third-party disability many SOs of individuals with tinnitus experience. The results indicate that SOs may benefit from a shared intervention to help mitigate the negative effects through a better understanding of tinnitus.

## 1. Introduction

Tinnitus is common, with at least one in 10 adults experiencing tinnitus [1]. Tinnitus can lead to many difficulties, including trouble sleeping, concentrating, and interference with the ability to hear [2,3]. People with tinnitus may reduce their working hours, and their mood may be significantly affected [4]. Some challenges also affect those they live with, as people severely affected by tinnitus may reduce socializing or household tasks they had previously participated in, due to fear of negatively affecting the tinnitus [5]. Such an effect on significant others (SOs) of individuals with disabilities and/or chronic health conditions experiencing indirect negative consequences as a result of their family members health condition is referred to as “third-party disability” according to the World Health Organizations (WHOs)—International Classification of Functioning, Disability, and Health (ICF) [6,7].

Research on chronic conditions has shown that various conditions, including several communication disorders, result in third-party disabilities for SOs [6,7,8,9,10,11]. Studies have highlighted the impact of hearing loss in terms of collateral psychosocial effects on SOs, including spouses, close family members, or caregivers [12]. These include auditory and social effects resulting from effort and fatigue [13]. SOs also need to go through a period of adjustment to adapt to the diagnosis of hearing loss of close family members or friends [9]. The hearing loss diagnosis can also change their perceptions of these people [14]. They furthermore need to assist individuals with hearing loss to manage everyday life activities such as group conversations [11]. Vestibular disorders may also have significant negative effects on SOs [15], and many SOs describe their experiences from coping to victimization [10]. As individuals with tinnitus experience various psychological and social consequences, it is reasonable to assume that their SOs may feel some of these effects on themselves.

As most of the focus is placed on the individual with tinnitus, not many studies have explored the impact of tinnitus on SOs. A few studies have examined the role of spouse responses and marital satisfaction in moderating the perceived severity of tinnitus and the experiences of emotional distress [16,17]. A study by Sullivan et al. [17] identified that poor marital cohesion, those with depression and highly critical spouses showed poor habituation to tinnitus. Another study by Pugh et al. [16] examined the contribution of spouse responses to tinnitus in 91 individuals with tinnitus and 74 spouses. Their results suggested that marital dysfunction and ignoring or selectively punishing complaints were related to anxiety and depression and mediated maladaptive coping and tinnitus severity. More recently, Mancini et al. [18,19] investigated responses from 41 individuals with tinnitus and 31 SOs and identified that from this sample, SOs did not always appreciate the difficulties experienced by those with tinnitus and that they had less knowledge about tinnitus and were thus not able to help. This study also reported that the SOs and their partners seldom spoke about tinnitus, which could result in misunderstandings regarding tinnitus and its consequences [19]. Although these studies have been helpful, the results need to be interpreted within the context of the samples used and biases thus arising, and further studies are required to explore these results.

Before identifying how to inform SOs about the effects of tinnitus, more information is first required on how tinnitus affects them. Therefore, the primary aim of the current study was to explore the effects of tinnitus on SOs by asking open-ended questions to qualitatively explore the impact. In addition, we also examined associations between SOs reporting of effects and tinnitus severity as rated by the individual with tinnitus.

## 2. Materials and Methods

### 2.1. Study Design

The study used a cross-sectional design and a mixed-methods approach for data collection and analyses. Ethical approval was obtained from the Institutional Review Board at Lamar University, Beaumont, Texas, USA (IRB-FY20-200).

### 2.2. Participants

Individuals with bothersome tinnitus who participated in trials of Internet-based cognitive behavioral therapy (ICBT) for tinnitus [20,21,22] were asked to consent to their SOs being involved in the study. When this was provided, the individual with tinnitus could pass on the study registration details to their selected SO. The inclusion criteria were providing informed consent and being a SO to an individual with tinnitus on the study. The exclusion criteria were not completing the open-ended questions on the SO questionnaire. As this was an exploratory study, the range of possible SO was broad and included a spouse, partner, parent, child, sibling, other family members, or a close friend. The role of the SOs was to complete a questionnaire regarding the effects of tinnitus on people that are close to those with tinnitus.

### 2.3. Data Collection

Online questionnaires were used throughout the study. Demographic information included gender, age, relation to the person with tinnitus, whether they had tinnitus themselves, and whether they lived with the person with tinnitus. Individuals with tinnitus completed the Tinnitus Functional Index (TFI) [23] as a measure of tinnitus severity. As the effect of tinnitus on SOs has received little attention, this study sought to identify the impact by asking broad questions so that the SOs were able to provide their own opinions. The following three open-ended questions were asked:
(i)Please describe in what ways the person’s tinnitus affects you.(ii)Please describe in what ways the person’s tinnitus has affected your relationship.(iii)Please list any positive experiences you have had as a result of your partner/family member/friend having tinnitus.

### 2.4. Data Analysis

Qualitative data were analyzed using qualitative content analysis [24]. The coding was independently performed using Excel spreadsheets and NVivo 12 software by EB and AU. The statements were initially read and re-read in search of initial ideas, meaning, and patterns. Patterns were then identified, and the data were coded. The responses that related to the same category were grouped. The codes were derived using a deductive approach from the key categories in the questionnaire. The smallest unit used for coding was phrases out of the open-ended statements. This process was performed separately for each of the three open-ended questions (the personal effect, the effect on the relationships, and positive experiences).

Statistical Package for Social Sciences (IMB SPSS for Windows v.26.0) was used for statistical analyses. Descriptive statistics including age, gender, and the relationship between the SO and individuals with tinnitus were used to describe the sample characteristics for each group. Continuous variables were summarized with means and standard deviations. Categorical variables were described using frequencies and percentages. Where ordinal data (the individual Likert scale questions) were present, the median was reported. When the scores from questions were combined (total scores), the mean scores were reported. Analysis of variance (ANOVA) and Chi-square analysis were performed to identify any group differences regarding baseline characteristics between those with and without positive experiences. A *p* value of 0.05 was used for significance interpretation, although a more stringent *p* value of 0.001, adjusted for multiple comparisons, was used where applicable.

## 3. Results

### 3.1. Significant Other Participants

There were 156 SOs who completed the open-ended questions. While this most likely constituted roughly 60% of SOs of the individuals with tinnitus that registered to undertake the ICBT, we were unable to estimate the response rate as we did not know how many individuals with tinnitus passed on the questionnaire to their SOs (including that not all may have had SOs close enough to ask). Table 1 provides the demographic characteristics of the SOs. Just over half of the SOs were male (53%), as seen in Table 1. Some of the SOs also reported tinnitus (*n* = 29; 19%). The majority of SOs were partners (84%) of individuals with tinnitus, although a small number of parents, children, other relatives, and friends also participated. Significant others were subdivided into three groups, namely those reporting that the other person’s tinnitus affected them personally (*n* = 150), their relationship (*n* = 152), and those who reported positive experiences associated with the other person’s tinnitus (*n* = 135), as shown in Table 1. When comparing the demographic characteristics of those reporting that the other person’s tinnitus affected them compared to not affecting them, no significant differences were found (see Table 2).

### 3.2. Individuals with Tinnitus

The mean age of the individuals with tinnitus was 57.14 years (SD: 11.56) (range: 21–81), and 58 (37%) were male and 98 (63%) were female. The mean Tinnitus Functional Index score was 52 out of 100 (SD: 20.21), range: 27–96, suggesting that the individuals with tinnitus had tinnitus severity that was severe enough to warrant a tinnitus intervention. The mean tinnitus severity was the same (57/100) for the individuals with tinnitus for each subgroup of SOs (those reporting an impact of the tinnitus on them personally, their relationship, and positive effects)

### 3.3. The Effects of Tinnitus on SOs

Responses from 127/150 (85%) SOs identified at least one impact of tinnitus. These responses fell into five categories and 20 subcategories (see Table 3, Figure 1). The impact included adjustments to the sound environment, activity limitations imposed by tinnitus, demand imposed by tinnitus, the emotional toll of tinnitus, and the resulting sense of helplessness. There was no association between significant others reporting an effect of tinnitus and the nature of relationship with the individual with tinnitus (e.g., partner, child, or the level of tinnitus severity (see Table 2).

For the remaining *n* = 23/150; (15%) of SOs, tinnitus had little effect as indicated by statements such as “Not very many effects. I just need to be thoughtful when I want to play music or news and simply be considerate about their feelings.”

### 3.4. The Effect of Tinnitus on Relationships

The majority of the responses (*n* = 92/158; 58%) indicated that tinnitus negatively impacted the relationship, mentioning the tinnitus is causing them to grow apart as they were less connected often due to withdrawal from the person with tinnitus, the effect of low mood from the individual with tinnitus, and worrying about the relationship. Communication was often mentioned to be strained or limited, mentioned by 57 (38%) participants. These difficulties appeared to be related to difficulties with tinnitus, the effect of tinnitus on levels of irritation, and/or hearing loss, and tinnitus. SOs described how restrained and frustrating communication was, resulting in many misunderstandings and withdrawals from conversations both by individuals with tinnitus and their SOs. When comparing the type of relationship (e.g., being the partner, parent, child, etc.) of those reporting that the other person’s tinnitus affected them compared to not affecting them, no significant differences were found (see Table 2).

The remaining 49 (42%) of SOs indicated that tinnitus did not have a negative impact with statements such as “not much as we have a strong relationship” or even strengthened their relationship as it resulted in them spending more time together, having more respect for each other, becoming closer, and having more open communication (see Table 4, Figure 2).

### 3.5. Positive Experiences as Result of Their Significant Others’ Tinnitus

When asked the question “Please list any positive experiences you have had as a result of your partner/family member/friend having tinnitus”, of the 135 that responded, *n* = 71 (53%) indicated that there were no positive effects, and *n* = 64 (47%) reported positive effects (see Table 5, Figure 3). For the majority, only one positive effect was mentioned. Of those reporting positive effects, 78% were partners and 8% were children. The association between SO status (partner, child, relative) and the number of positive experiences was not statistically significant, and neither was the presence of tinnitus for significant others. The mean TFI score of the individuals was higher (*p* = 0.02) in the group reporting positive experiences (57, SD: 21) than the group not reporting positive experiences (53 SD: 19) but this association was not significant.

Qualitative analysis indicated that the positive experiences included lifestyle adaptions, personal development, health awareness, and a changed outlook (see Table 5).

## 4. Discussion

As there are few studies exploring the impact of tinnitus on SOs, this exploratory study was undertaken. The study investigated the effects of tinnitus on 156 SOs and the relationship between SOs and those with tinnitus. It furthermore sought to identify any positive effects of tinnitus on SOs. The findings and their implications are discussed below.

### 4.1. The Effect of Tinnitus on SOs

The majority (85%) of significant others reported that tinnitus impacted them, indicating that SOs may experience third-party disability. This supports the finding regarding third-party disability for other disabilities such as aphasia [6], hearing loss [7], and vestibular difficulties [15]. The impact surrounded sound adjustments, activity limitations, additional demands, emotional toll, and helplessness. The use of sound enrichment is a frequently used strategy to reduce tinnitus perception. Having to live with constant background noise during the day and night-time was particularly difficult for SOs to adapt to. Furthermore, many SOs mentioned difficulties in constantly having to regulate sound, as some things were too loud and other things too quiet for those with tinnitus. They also found that due to probable associated hearing difficulties, often going hand-in-hand with tinnitus, appliances such as the television and music players were very loud, which was unpleasant for them. Audiologists should be mindful of these possible effects on SOs and identify ways to address these difficulties.

SOs also found that due to fear of tinnitus being exacerbated, activities they had previously participated in were now limited. This included attending fewer social functions and concerts and listening to music less. The difficulty for SOs of having to adapt to these changes and reduction of social activities has also been found for SOs of those with aphasia [8] and hearing loss [9,12,13]. Going out is not as easy, as they need to be mindful of the impact of the weather and other factors. Ways of helping those with tinnitus overcome these difficulties to be able to improve their quality of life are important. Interventions for people with bothersome tinnitus which incorporate exposure to feared situations may be helpful [25].

It was insightful that due to being the SO of someone with tinnitus, they found this role demanding and emphasized the negative impact. SOs of those with Meniere’s disease were found to vary between those not being affected to some feeling victimized [10]. The fatigue and need to assist have also been mentioned by SOs of those with hearing loss [11,13]. This impact is often not considered, and more should be done to support SOs. Incorporating them into tinnitus interventions should be considered [19]. They also mentioned feeling helpless in their endeavors to support those with tinnitus. Such reports of people without tinnitus, especially the SOs not being able to understand what it is like living with tinnitus, has been reported by individuals with tinnitus in their online conversations [26]. Ways of improving their understanding of tinnitus and what may be helpful should be sought. This suggests that some form of combined interventions may be helpful and should involve SOs [19].

### 4.2. The Effects of Tinnitus on Relationships

Tinnitus negatively affected the relationships for 58% of SOs. In particular, communication was often negatively affected, as it was more limited due to tinnitus and hearing difficulties, as previously reported for SOs with hearing loss [9]. These difficulties strained communication, resulting in frustrations and misunderstandings. SOs also felt they had grown apart as they were less connected. This was partly due to the effect of the person’s with tinnitus mood on interactions, both partners withdrawing and worrying about being able to deal with these effects in the long term. This may have the effect of perceptions of each other changing within the relationship, as has been found for hearing loss, which may negatively impact the relationship further [14]. It has previously been suggested that tinnitus habituation is poorer when SOs are perceived as more critical [17]. The way SOs respond to tinnitus furthermore affected coping, as also suggested by Pugh et al. [16]. Many SOs felt they were growing apart, however, some SOs mentioned that they were spending more time together as the person with tinnitus relied more on them. This relates to previous research indicating the importance of family support and attachment for those with tinnitus [27,28]. Respect had also been earned for the way the individual with tinnitus handled it. Tinnitus also resulted in more open discussions about issues, which had a positive impact on relationships. Such positive effects on relationships have been reported by SOs of individuals with hearing loss and Meniere’s disease [29]. These effects again indicate that involving SOs in tinnitus intervention may be beneficial. Internet intervention for tinnitus [30,31] could, for instance, be adapted to include modules for SOs, as has been done for other disorders [32,33].

### 4.3. Positive Experiences

Almost half (46%) of the SOs reported some positive experiences related to tinnitus, including lifestyle adaptions such as engaging in more relaxing activities, partaking in new past times, and slowing down the pace of time. Personal development was also found due to developing more empathy, patience, and understanding for those with tinnitus. They also became more health-aware, resulting in healthier lifestyles, use of hearing protection, and being aware of hearing loss. Positive experiences have been reported both by individuals affected and SOs in hearing loss and Meniere’s disease [29]. Tinnitus awareness also led to a changed outlook through increased knowledge due to investigating tinnitus and being more appreciative and grateful. Similar themes were also found with regard to hearing impairments [34]. These SOs also mentioned developing patience and tolerance, understanding and awareness of hearing problems, improved communication skills, and doing activities that do not rely on hearing. They reported that children and grandchildren were more likely to report positive experiences than partners. This finding was not observed in the present study.

There was also some overlap between the positive effects mentioned by the SOs from the present study and those mentioned by individuals with tinnitus in a previous study [35]. Positive mentioned tinnitus effects for both SOs and those with tinnitus included personal development, gratefulness, lifestyle changes, awareness of hearing protection, a better understanding of tinnitus and a changed outlook. Tinnitus interventions should include elements where participants can identify possible positive effects for both them and SOs and help create opportunities for these.

### 4.4. Limitations and Future Directions

The participants represented SOs of those with bothersome tinnitus who felt they required an intervention to help them. They may thus not represent all individuals with tinnitus. Individuals who have more severe tinnitus are more likely to have passed on the questionnaire to their SOs. Further reporting bias to consider is that the SOs selecting to participate may be the ones noticing the effect. The SOs were also asked three leading questions which could have resulted in participants considering effects they had not previously considered. Thus, this sample may have reported more severe consequences as a result of their partner’s tinnitus. The confounding effect of hearing loss was also not accounted for in this sample. SO may have attributed negative effects of their communication partner’s hearing loss and tinnitus and not entirely to that of the partner’s tinnitus. The inclusion of SPs with tinnitus may have resulted in more severe effects being reported. An additional weakness was the lack of a control or comparison group. Future studies should make an effort to include a more representative sample of SOs. The consequences identified in this study were used to design a structured questionnaire for SOs to identify third-party disability for tinnitus [36]. This questionnaire can be used to identify the effects of undertaking tinnitus interventions on SOs.

## 5. Conclusions

This exploratory study identified that SOs perceived that tinnitus had an impact on them and their relationship with individuals with tinnitus. SOs of individuals with tinnitus indicated feeling helpless at both understanding the difficulties faced and how to help those with tinnitus, as also previously reported by Mancini et al. [18,19]. These reports highlighted the third-party disability experienced by SOs of individuals with tinnitus. It was, however, encouraging that positive experiences were reported by some SOs. Seeking positive experiences may help reduce the negative impact of tinnitus on SOs. Providing SOs with support and a better understanding of tinnitus may result in them assisting individuals with tinnitus better. This may mitigate some of the negative results of tinnitus on SOs and the relationship with the individual with tinnitus. At present, no such interventions exist, and considerations to this provision should be made.

## Figures and Tables

**Figure 1 jcm-11-01393-f001:**
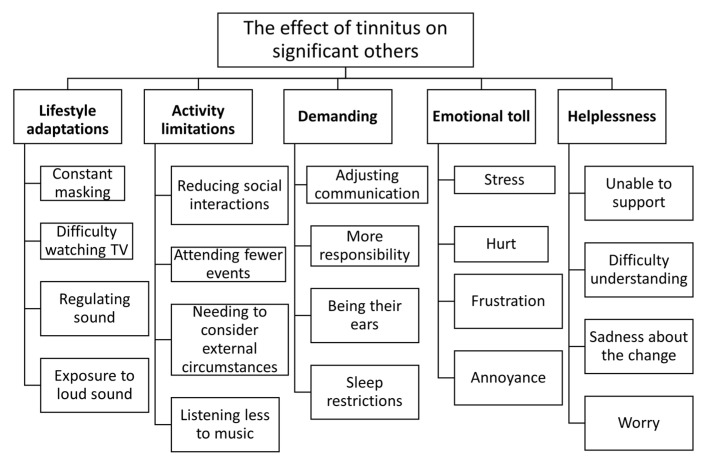
The effect of tinnitus on significant others.

**Figure 2 jcm-11-01393-f002:**
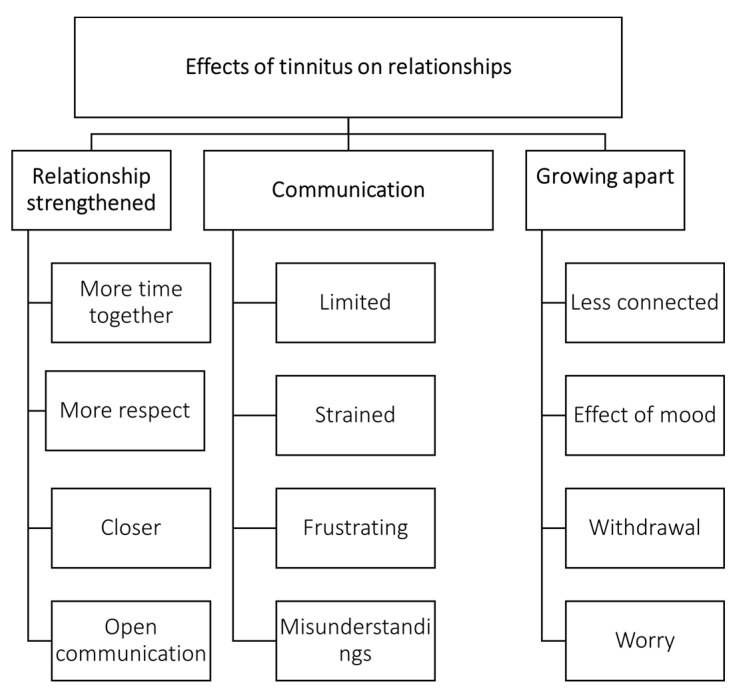
The effects of tinnitus on relationships.

**Figure 3 jcm-11-01393-f003:**
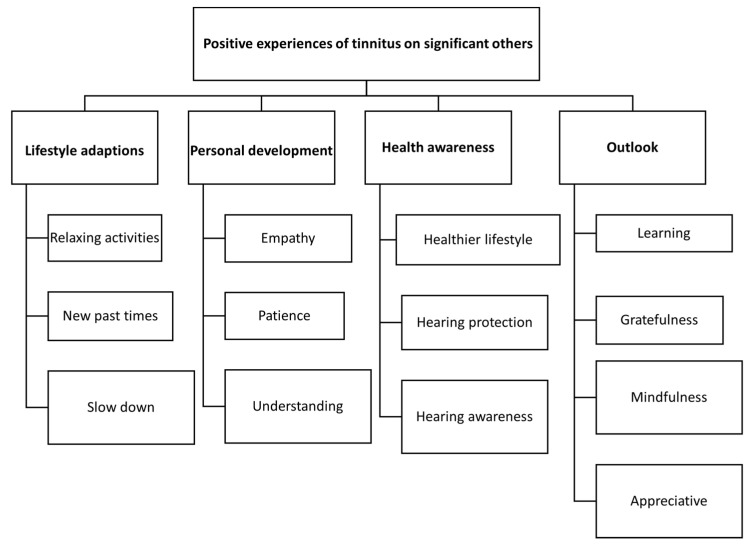
Positive experiences of tinnitus on significant others.

**Table 1 jcm-11-01393-t001:** Demographic profile of the significant others.

Characteristic	All Significant Others	Subgroups of Significant Others		
	(*n* = 156)	Significant Others Finding the Other Person’s Tinnitus Affects Them Personally (*n* = 150)	Significant Others Reporting the Other Person’s Tinnitus Affects Their Relationships (*n* = 152)	Significant Others Mentioning Positive Experiences from the Other Person’s Tinnitus (*n* = 135)
Demographics *n* (%)				
Mean age (Standard deviation) (Range)	55.96 (14.24) (18–84)	55.66 (14.37) (18–84)	55.61 (14.56) (19–84)	55.47 (14.69) (22–84)
Gender				
Male	82 (53%)	48 (38%)	47 (50%)	33 (52%)
Female	74 (47%)	79 (62%)	48 (50%)	31 (48%)
Relationship				
Partner	131(84%)	108 (85%)	81 (85%)	50 (78%)
Parent	2 (1%)	1 (1%)	0	2 (3%)
Child	11(7%)	7 (6%)	9 (10%)	5 (8%)
Relative	7 (5%)	6 (5%)	3 (3%)	4 (6%)
Friend	5 (3%)	5 (4%)	2 (2%)	3 (5%)
Living together *n* (%)				
Yes	135 (87%)	111 (87%)	82 (86%)	49 (77%)
No	20 (13%)	16 (13%)	13 (14%)	15 (23%)
Presence of tinnitus				
Yes	29 (19%)	24 (19%)	16 (17%)	13 (20%)
No	127 (81%)	104 (81%)	79 (83%)	51 (80%)

**Table 2 jcm-11-01393-t002:** The association between significant others’ characteristics and those reporting that tinnitus affected them.

Significant Others Reporting Effects (*n* = 156)	Significant Others Finding the Other Person’s Tinnitus Affects Them Personally (*n* = 150)	Significant Others Reporting the Other Person’s Tinnitus Affects Their Relationships (*n* = 152)	Significant Others Mentioning Positive Experiences from the Other Person’s Tinnitus (*n* = 135)
Nature of the relationship	*X*^2^ (4) = 6.8, *p* = 0.15	*X*^2^ (4) = 7.41, *p* = 0.12	*X*^2^ (4) = 2.99, *p* = 0.56
Living together	*X*^2^ (1) = 3.7, *p* = 0.06	*X*^2^ (1) = 0.06, *p* = 0.81	*X*^2^ (1) = 2.50, *p* = 0.11
Significant other experiencing tinnitus	*X*^2^ (1) = 0.03, *p* = 0.87	*X*^2^ (1) = 0.82, *p* = 0.37	*X*^2^ (1) = 0.2.93, *p* = 0.09
Level of tinnitus severity	*X*^2^ (4) = 6.8, *p* = 0.15	*X*^2^ (4) = 7.41, *p* = 0.12	*X*^2^ (4) = 2.99, *p* = 0.56
Mean TFI score for individuals with tinnitus for each group	57/100 (SD: 20)	57/100 (SD: 21)	57/100 (SD: 21)

**Table 3 jcm-11-01393-t003:** The impact of tinnitus on significant others.

Category	Subcategory	Number of Meaning Units (*n* = 238)	Example of Meaning Unit
Sound adjustments	Constant background sound	10	I tend to get angry because she always has to have something in the room creating background noise in order to sleep.
	Television watching	15	He misses stuff even on full volume and we have to watch parts of a TV show over and over and over.
	Regulating sound	16	He complains when I play loud music. I have to keep the house quiet. I wear Bluetooth earphones to watch TV so as not to bother him. I have to keep my voice down. If you met me, you would know how hard that is!
	Exposure to loud sound	10	She keeps the TV so loud I can hear it across the house with the door closed. The volume of the truck radio is painfully loud.
Activity limitations	Reducing social interactions	16	Our social life and time with our adult kids has been affected as well. When we are in a big family gathering, he will leave the room because he gets so anxious over the noise level. It is embarrassing. Or I go to the functions without him, which sucks.
	Attending fewer events	17	Greatly reduced our going out to eat or large gatherings where there is a lot of noise. We can’t go to concerts, theaters and movies due to the volume.
	Needing to consider external circumstances	11	It has affected our times together. Mostly at night when it is at its worst. It has kept us from going outside when it seems to be worse in humidity or in certain weather.
	Listening less to music	9	We listen to less music together. He dislikes most music because he can’t hear words. Greatly has changed our enjoyment and love of music.
	Total number of meaning units	104/238 (44%)	
Demanding	Adjusting communication	21	For someone to not be able to hear when you talk to them unless you are right in front of them is frustrating. It can cause you to not say things that you otherwise would’ve.
	More responsibility	16	I am shouldering more responsibility for our children and other duties.
	Being their ears	12	I am the one who is repeating back to him what he missed, and it can be very difficult to keep up, especially if you are watching a movie and also in conversation.
	Sleep restrictions	5	Constant sound together with sleep disruptions from my partner affects my daily life.
	Total number of meaning units	54 (23%)	
Emotional toll	Stress	9	I feel stressed out when she shows signs of depression or low mood.
	Hurt	5	He gets anxious when the tinnitus is bad. He will be short with me and speak harshly. Hurts. I feel helpless.
	Frustration	14	Becoming frustrated with the continued complaints of tinnitus. I tend to lose patience, especially if I have to repeat myself multiple times.
	Annoyance	17	She complains about how it affects her. We can become annoyed at each other.
	Total number of meaning units	45 (18%)	
Helplessness	Unable to support	8	My wife sometimes starts crying for no apparent reason. She often complains about the sounds and I have no way of helping her. I feel powerless to help my wife and this is a burden I cannot handle very well.
	Difficulty understanding	14	I find it hard to understand how difficult it is to live with this disease. In other words, it is impossible to feel what they are feeling. I do understand that it is a debilitating disease.
	Sadness about the change	5	It makes me feel very frustrated that I can’t help her and I’m sad that she is not the same person as before.
	Worry	8	I worry about her health and ability to cope with it. I worry about her not being able to read for very long since she is an avid reader.
	Total number of meaning units	35 (15%)	

**Table 4 jcm-11-01393-t004:** Tinnitus effects on the relationship between significant others and the individual with tinnitus.

Category	Subcategory	Number of Meaning Units (*n* = 282)	Example of Meaning Unit
Relationship strengthened	More respect	3	A greater level of respect for how he handles the tinnitus. I have hyperacusis and don’t handle it very well a great deal of the time.
	Brought us closer	10	We have in some ways grown closer because she does not like to be alone and relies on me more than before.
	Openness	3	He opens up about issues he is having
	Total number of meaning units	16/282 (6%)	
Communication difficulties	Limited	19	Believe me. I never know when I need to speak up for her to hear me or when I am speaking too loudly—until she lets me know in not always the gentlest fashion. This causes me to withdraw from trying to communicate.
	Strained	17	Daily communication is a chore. Our communication has suffered a great deal because he can’t concentrate on conversation and keeps asking for things to be repeated.
	Frustrating	19	I’m always yelling and get frustrated having to repeat myself for him to hear and understand me. As I have to raise my voice which perceived that I am angry or upset.
	Misunderstandings	25	We have had many communication errors. I have told him something important that is misheard. It is not just the misunderstandings but also the emotional effect. It has me mistakenly think he is mad or disgusted with me when he is mad and disguised at it.
	Total number of meaning units	80 (28%)	
Growing apart	Less connected	15	I feel he shuts me out because he is scared and doesn’t want to scare me or look “weak.” I have walked on eggshells and am afraid to set him off. Our level of connection has diminished somewhat.
	Effect of mood	16	I often feel like I have to “walk on eggshells” around her and be careful not to upset her. Because she is often on edge. She gets irritable more readily and overwhelmed easily. Mostly the low mood portion is what impacts our relationship She can get grumpy and quick tempered.
	Withdrawal	8	It has been tough to cope with the illness and the misery that my wife is going through. We have been limited in the activities that we use to do together before the tinnitus and she now tends to just want to be by herself.
	Worry	6	It has concerned both of us as to how we will face it in the future as we both know it will surely get worse.
	Total number of meaning units	45/282 (16%)	

**Table 5 jcm-11-01393-t005:** Positive experiences of tinnitus on significant others.

Category	Subcategory	Number of Meaning Units (*n* = 88)	Example of Meaning Unit
Lifestyle	Relaxing activities	4	I have spent more reflective time rather than just having noise going in the house with the TV etc.
	New pastimes	7	I cook more at home.
	Slow down	4	When it’s bad we stop slow down do and work on it.
		15/88 (17%)	
Personal development	Empathy	10	I’ve learned to be more empathetic.
	Patience	15	Taught me to be more patience and caring.
	Understanding	13	The only positive thing would be the awareness of tinnitus and how many people suffer with it. I am more aware of it and other people I come across who have it.
		38/88 (44%)	
Health awareness	Healthier lifestyle	10	Our eating habits have improved with hopes of relieving the symptoms of tinnitus.
	Hearing loss awareness	9	I now wear earplugs at concerts.
	Investigations	6	It has led me to try to help do some research with him on possible remedies. We have had some positive discussions on what he should do when it seems really bad.
		25/88 (28%)	
Outlook	Gratefulness	10/88 (11%)	I’m more gratitude for simple things.

## Data Availability

All data relevant to the study are included in the article. Due to the nature of this research, participants of this study did not agree for their interview transcripts or recordings to be shared.

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
