# Peer review of "The Effects of Tinnitus on Significant Others"

_jcm, 2022, doi:10.3390/jcm11051393_

Round 1

Reviewer 1 Report

The Effects of Tinnitus on Significant Others

Brief Summary

The aim of the study was to evaluate the impact of tinnitus on significant others (SO) of individuals suffering from this condition. The Tinnitus Functional Index (TFI) was administered on individuals with tinnitus, while SO answered three open-ended questions about the impact of tinnitus. Results revealed that 85% of SO were impacted by tinnitus. 58% reported a negative impact while 47% reported positive impact (not mutually exclusive). The authors propose a shared intervention for both individuals with tinnitus and their SO to mitigate the impact of tinnitus.

Introduction:

  • Please address comments in the pdf file. Otherwise, the introduction is well written and to-the-point. This is a relevant and timely study.

Methodology:

  • 2 Please define ‘large sections’ under exclusion criteria
  • Please elaborate on the strategy used for content coding. Some methodological details are not clear. For instance:
    • What was the smallest unit used for coding – words, phrases, sentences?
    • In Tables 2 and 4, are the categories and subcategories mutually exclusive? How were they created?
    • It would be helpful to know the total number of meaning units to put the proportion of each category/subcategory in context.
    • One of the examples in Table 2 “He gets anxious when the tinnitus is bad. He will be short with me and speak harshly. Hurts. I feel helpless. Frustrated. I think he is mad at me, but he is really probably just scared and frustrated.” is classified under the sub-category ‘Hurt’. It could have been easily classified under the sub-category of ‘Frustration’. Another example: “She complains about how it affects her. We can become annoyed at each other. Tired of repeating things over and over. We often disagree on the volume of the TV which can make the experience unpleasant for one of us.” This could have been under the category of ‘Sound Adjustment’. It would be helpful for the reader to understand what procedures were used to categorize results and to resolve conflict.
    • Similarly in Table 4, “Our level of connection has diminished” is classified under ‘Less connected’, but “I feel he shuts me out because he is scared and doesn't want to scare me or look “weak.” I have walked on eggshells and afraid to set him off. Our level of connection has diminished somewhat.” is categorized under ‘Withdrawal’

Results:

  • It would be interesting to explore whether the proximity of relationship (whether living together) affected results
  • This is minor, but I think Figures 2 and 3 may have switched

Discussion:

  • The discussion section needs to be more fleshed out, especially discussing the implications of the results rather than repeating the findings. Since there are limited previous studies on tinnitus, studies of the effects of other health conditions on SO may need to be incorporated into the discussion.

Miscellaneous:

  • Please proof-read the manuscript for typographical errors and grammatical accuracy before resubmission – some sentences that need attention are highlighted.

Line-by-line Comments:

  • Additional comments are provided as annotations and ‘sticky notes’ in the attached .pdf file.

Author Response

The Effects of Tinnitus on Significant Others

Reviewer 1

The authors studied the possibility of tinnitus causing third-party disability. It was an important study as the third-party is always being sidelined in the management of the patient.

Authors response:

Many thanks for the time taken to review this manuscript and for the helpful comments. The suggestions have been addressed as indicated below.

Comments:

1. To state the inclusion and exclusion criteria for the study

Authors response:

These are stated in section 2.2. under Participants as follows:

The inclusion criteria was providing informed consent and being a SO to an individual with tinnitus on the study. The exclusion criteria was not completing the open-ended  questions on the SO questionnaire.

2. To state the reason of using open ended question in the study. Certain important aspects might be missed.

Authors response:

This has been clarified in section 2.3 Data Collection

As the effect of tinnitus on SOs has received little attention this study sought to identify the impact by asking broad questions so that SO’s were able to provide their own opinions.

The results of these open-ended questions were used to design a self-reported questionnaire as stated in the discussion under section 4.4:

The consequences identified in this study were used  to design a structured questionnaire for SOs to identify third-party disability for tinnitus [36]. This questionnaire can be used to identify the effects of undertaking tinnitus interventions on SOs.

3. The inclusion of SOs with tinnitus may give a “more severe” result.

Authors response:

This has been added as a limitation as follows:

The inclusion of SPs with tinnitus may have resulted in more severe effects being reported.

4. The effect of tinnitus to the different groups of SOs (spouse / partner / parent / child etc) may be different.  Each group should be assessed separately.

Authors response:

This was done, and no significant differences were found as described in Section 3.4 The effect of tinnitus on relationships:

When comparing the type of relationship (e.g. being the partner, parent, child, etc) of those reporting the other person’s tinnitus affected them compared to not affecting them, no significant differences were found (see Table 2).

Reviewer 2 Report

The authors studied the possibility of tinnitus causing third-party disability. It was an important study as the third-party is always being sidelined in the management of the patient.

Comments:

  1. To state the inclusion and exclusion criteria for the study
  2. To state the reason of using open ended question in the study. Certain important aspects might be missed.
  3. The inclusion of SOs with tinnitus may give a “more severe” result.
  4. The effect of tinnitus to the different groups of SOs (spouse / partner / parent / child etc) may be different.  Each group should be assessed separately.

Author Response

The Effects of Tinnitus on Significant Others

Reviewer 2 

Brief Summary

The aim of the study was to evaluate the impact of tinnitus on significant others (SO) of individuals suffering from this condition. The Tinnitus Functional Index (TFI) was administered on individuals with tinnitus, while SO answered three open-ended questions about the impact of tinnitus. Results revealed that 85% of SO were impacted by tinnitus. 58% reported a negative impact while 47% reported positive impact (not mutually exclusive). The authors propose a shared intervention for both individuals with tinnitus and their SO to mitigate the impact of tinnitus.

 Authors response:

Many thanks for the time taken to review this manuscript and for the helpful comments. The suggestions have been addressed as indicated below.

Introduction:

  • Please address comments in the pdf file. Otherwise, the introduction is well written and to-the-point. This is a relevant and timely study.

Authors response:

 Thank you very much for taking the trouble to highlight these errors in the pdf file. They have all been addressed.

Methodology:

  • 2 Please define ‘large sections’ under exclusion criteria

Authors response:

This has been better defined as:

The exclusion criteria was not completing the open-ended questions on the SO questionnaire.

  • Please elaborate on the strategy used for content coding. Some methodological details are not clear. For instance:
    • What was the smallest unit used for coding – words, phrases, sentences?

Authors response:

This has been added to section 2.4 Data Analysis

The smallest unit used for coding were phrases out of the open-ended statements

    • In Tables 2 and 4, are the categories and subcategories mutually exclusive? How were they created?

Yes, they were mutually exclusive as they were based on different open-ended questions. This has been clarified in section 2.4 as follows:

Authors response:

This process was performed separately for each of the three open-ended questions (the personal effect, the effect on relationships and positive experiences).

    • It would be helpful to know the total number of meaning units to put the proportion of each category/subcategory in context.

Authors response:

The total numbers have been added to table 3- 5.

    • One of the examples in Table 2 “He gets anxious when the tinnitus is bad. He will be short with me and speak harshly. Hurts. I feel helpless. Frustrated. I think he is mad at me, but he is really probably just scared and frustrated.” is classified under the sub-category ‘Hurt’. It could have been easily classified under the sub-category of ‘Frustration’.

Authors response:

The first part of this sentence was used for the category ‘hurt’ and the second part for the category frustration. We have hence removed the second part of this sentence.

    • Another example: “She complains about how it affects her. We can become annoyed at each other. Tired of repeating things over and over. We often disagree on the volume of the TV which can make the experience unpleasant for one of us.” This could have been under the category of ‘Sound Adjustment’. It would be helpful for the reader to understand what procedures were used to categorize results and to resolve conflict.

Authors response:

The first part of this sentence was used for the category ‘annoyance’ and the second part for the category sound adjustment. We have hence removed the second part of this sentence.

    • Similarly in Table 4, “Our level of connection has diminished” is classified under ‘Less connected’, but “I feel he shuts me out because he is scared and doesn't want to scare me or look “weak.” I have walked on eggshells and afraid to set him off. Our level of connection has diminished somewhat.” is categorized under ‘Withdrawal’

 Authors response:

We have used this example under less connected and a new example for withdrawal as follows:

It has been tough to cope with the illness and the misery that my wife is going through. We have been limited in the activities that we use to do together before the tinnitus and she now tends to just want to be by herself.

Results:

  • It would be interesting to explore whether the proximity of relationship (whether living together) affected results

Authors response:

This has been added to table 2 and is non-significant.

  • This is minor, but I think Figures 2 and 3 may have switched

Authors response:

Thank you for pointing this out. These have been corrected.

Discussion:

  • The discussion section needs to be more fleshed out, especially discussing the implications of the results rather than repeating the findings. Since there are limited previous studies on tinnitus, studies of the effects of other health conditions on SO may need to be incorporated into the discussion.

 Authors response:

Thank you, we have amended the discussion section according to these suggestions.

Miscellaneous:

  • Please proof-read the manuscript for typographical errors and grammatical accuracy before resubmission – some sentences that need attention are highlighted.

Authors response:

Thank you, we have proof-read the manuscript and corrected the errors.

Line-by-line Comments:

  • Additional comments are provided as annotations and ‘sticky notes’ in the attached .pdf file.

Round 2

Reviewer 2 Report

The manuscript has been sufficiently improved.